# Effects of Different Continuous Aerobic Training Protocols in a Heterozygous Mouse Model of Niemann-Pick Type C Disease

**DOI:** 10.3390/jfmk5030053

**Published:** 2020-07-18

**Authors:** Ida Cariati, Manuel Scimeca, Virginia Tancredi, Agata Grazia D’Amico, Gabriele Pallone, Mattia Palmieri, Claudio Frank, Giovanna D’Arcangelo

**Affiliations:** 1Medical-Surgical Biotechnologies and Translational Medicine, Department of Clinical Sciences and Translational Medicine, “Tor Vergata” University of Rome, Via Montpellier 1, 00133 Rome, Italy; ida.cariati@uniroma2.it; 2Department of Biomedicine and Prevention, “Tor Vergata” University of Rome, Via Montpellier 1, 00133 Rome, Italy; manuel.scimeca@uniroma2.it; 3Department of System Medicine, “Tor Vergata” University of Rome, Via Montpellier 1, 00133 Rome, Italy; gabriele.pallone@gmail.com (G.P.); mattia.palmieri@live.it (M.P.); giovanna.darcangelo@uniroma2.it (G.D.); 4Centre of Space Bio-medicine, “Tor Vergata” University of Rome, Via Montpellier 1, 00133 Rome, Italy; 5Department of Drug Science, University of Catania, 95125 Catania, Italy; agata.damico@unict.it; 6National Center for Rare Diseases, Istituto Superiore di Sanità, viale Regina Elena 299, 00161 Rome, Italy; claudio.frank@iss.it

**Keywords:** aerobic training, neurodegeneration, mouse model, synaptic plasticity, muscle plasticity, Niemann-Pick type C disease

## Abstract

The positive effects of physical activity on cognitive functions are widely known. Aerobic training is known to promote the expression of neurotrophins, thus inducing an increase in the development and survival of neurons, as well as enhancing synaptic plasticity. Based on this evidence, in the present study, we analyze the effects of two different types of aerobic training, progressive continuous (PC) and varying continuous (VC), on synaptic and muscular plasticity in heterozygous mice carrying the genetic mutation for Niemann-Pick type C disease. We also analyze the effects on synaptic plasticity by extracellular recordings in vitro in mouse hippocampal slices, while the morphological structure of muscle tissue was studied by transmission electron microscopy. Our results show a modulation of synaptic plasticity that varies according to the type of training protocol used, and only the VC protocol administered twice a week, has a significantly positive effect on long-term potentiation. On the contrary, ultrastructural analysis of muscle tissue shows an improvement in cellular conditions in all trained mice. These results confirm the beneficial effects of exercise on quality of life, supporting the hypothesis that physical activity could represent an alternative therapeutic strategy for patients with Niemann-Pick type C disease.

## 1. Introduction

Niemann-Pick type C Disease (NPCD) is an autosomal recessive lysosomal storage disorder with progressive neurodegeneration, characterized by the insurgence of several visceral, as well as neurological symptoms. This rare disease is caused by an NPC1 and/or NPC2 gene mutation. Progressive neurological deterioration and insurgence of symptoms like ataxia, seizure and cognitive decline until severe dementia are observed. Speech impairment (such as slurring), swallowing problems, and psychiatric disorders may develop. Other symptoms may include sudden loss of muscle strength, which may vary from head nodding to complete collapse, abnormal posturing of the limbs, and lung complications [1].

The degeneration of cerebellar Purkinje cells, hippocampal and cortical neurons, hyperphosphorylated Tau and neurofibrillary tangle formation with the accumulation of lipid storage bodies are the neuropathological features of NPC1 brains [2,3,4,5,6].

It is commonly believed that these neurological symptoms do not develop in heterozygous carriers of NPC1 mutations; nonetheless, it has been shown by Kluenemann et al. [7] that three adult heterozygous carriers of mutations in the NPC1 gene also had a Parkinsonism syndrome. This comorbidity suggests the possibility that mutations in NPC1 could be a risk factor for Parkinson’s disease similar to the phenomenon that is now recognized with Gaucher disease and the glucocerebrosidase (GBA) gene.

Beneficial effects of physical exercises on the nervous system from neuroprotection to neuronal plasticity are reported [8,9,10].

Intervention studies of both humans and rodents have indicated that mental decline resulting from aging and neurodegenerative diseases is ameliorated by sustained exercise [11]. Furthermore, it is thought that physical activity may slow down Multiple Sclerosis pathology course [12], decrease the risk of dementia [13], and prevent the onset of Parkinson’s disease and improve its symptoms [14].

The results from our laboratories [10] indicate an impairment of long-term potentiation (LTP) recorded in hippocampal slices both in NPC1 −/− and NPC1 +/− sedentary mice, which showed partial recovery after a short training period of uniform continuous (UC) training only in the homozygous animals. LTP is the electrophysiological paradigm of learning and memory and is impaired in compromised cognitive processes.

To mimic conditions of higher risk for neurodegenerative diseases in heterozygous human patients, in the present paper we evaluated hippocampal neuroplasticity of NPC1 +/− in continuously trained mice, with both progressive and varying training, by applying electrophysiological techniques. In addition, we evaluated the effects of these different training activities on muscle ultrastructure by transmission electron microscopy analysis.

## 2. Materials and Methods

### 2.1. Animals

We used 16 one-month-old BALB/c mice carrying the genetic mutation for NPC1 (BALB/cNctr-Npc1^m1N^/J). We first established a colony of NPC1 −/− mice (Stock number: 003092) purchased from Jackson Laboratories (Bar Harbor, MA, USA). We bred heterozygous mice, and the genotypes of offspring animals were determined as indicated by Jackson Laboratories in the genotyping protocols database using polymerase chain reaction (PCR) [15]. Experiments were performed after approval of the project by the Ethical Committee of the University of Rome “Tor Vergata” (approval n. 86/2018-PR) and according to the procedures established by the European Union Council Directive 2010/63/EU for animal experiments [16]. The animals were divided into three groups (four mice per group), each one submitted to different aerobic training programs, and a fourth control group (four mice), which did not perform any type of training.

### 2.2. Behavioral Training

A Rotarod (Cat N° 47600, Ugo Basile srl, Milan, Italy) was employed for the administration of the various aerobic training protocols and for the evaluation of their effects. Before starting the training program, the mice underwent an incremental test, to establish the volume of work that could be sustained with a maximum daily number of three falls from the Rotarod, as previously reported [17]. We administered two aerobic training protocols, progressive continuous (PC) and varying continuous (VC), which differ in terms of speed and speed variations, as described in our previous work [17]. The PC protocol consisted of 18 min of training at a gradual speed of rotations increasing from low to high intensity (10–32 RPM). The CV protocol was characterized by two 8-min reverse bipyramidal series, with two minutes of active recovery between series at 10 RPM, for a total duration of 18 min. Training sessions were conducted three times a week for 12 weeks (PCt and VCt), and in the case of the VC protocol, also twice a week (VCb). During each training session, the number of falls of the mice from the Rotarod was evaluated. Moreover, each mouse was weighed before and after training.

### 2.3. Electrophysiological Recordings in Mouse Hippocampal Slices

At the end of the training period, the mice were sacrificed as previously reported [18]. The obtained hippocampal slices were transferred to an interface tissue chamber to record long-term potentiation (LTP) and evaluate how aerobic training could affect this synaptic plasticity phenomenon, characterized by increasing intensity of synaptic transmission and considered the electrophysiological paradigm of learning and memory processes [19]. The extracellular recordings of population spike (PS), that indicates the electrical activity of a population of neurons, and the subsequent analyses were made according to procedures previously indicated [20].

### 2.4. Transmission Electron Microscopy (TEM)

One mm^3^ of muscle tissue from surgical specimens were fixed in 4% paraformaldehyde and post-fixed in 2% osmium tetroxide [21]. After washing with 0.1 M phosphate buffer, the sample was dehydrated by a series of incubations in 30%, 50%, and 70% ethanol. Dehydration was continued by incubation steps in 95% ethanol, absolute ethanol and propylene oxide, after which samples were embedded in Epon (Agar Scientific, Stansted, Essex CM24 8GF United Kingdom) [22]. Eighty nm ultra-thin sections were mounted on copper grids and examined with a transmission electron microscope (Model JEM-1400 series 120 kV, JEOL USA, Inc. 11 Dearborn Road Peabody, MA 01960; DigitalMicrograph^TM^ Software).

### 2.5. Statistical Analysis

Statistical analysis was performed using GraphPad Prism 8 Software (Prism 8.0.1, La Jolla, CA, USA). For electrophysiological experiments, data were expressed as mean ± SEM and *n* represents the number of slices analyzed. Data were compared with ANOVA and Tukey’s Multiple Comparison Test and were considered significantly different if *p* < 0.05.

## 3. Results

### 3.1. Effects of Different Training Protocols on Physical and Performance Parameters

Mice weight was measured throughout the entire experimental procedure, and is shown in Figure 1A. Before the start of the training sessions (at one month of life), the mice belonging to each experimental group had a similar mean body weight (CTRL 18.5 ± 0.4 g, PCt 17.5 ± 0.5 g, VCt 19.7 ± 0.3 g, VCb 21.3 ± 0.7 g). We observed that the three different training protocols induced a significant increase in mean body weight in each of the groups of trained mice (PCt 27.7 ± 1.0 g, VCt 28.7 ± 0.7 g, VCb 31.0 ± 0.6 g, **** *p* < 0.0001), while we found no significant differences in the control group (20.5 ± 0.4 g). In particular, the increase in mean body weight after training was 58% in the PCt group, 46% in both VC groups and only 11% in the CTRL group.

During the 12 weeks of training, we also evaluated the number of falls of the mice from the Rotarod for every type of protocol (Figure 1B). The mean number of falls during the training period were—131.2 ± 27.7 in the PCt group, 420.0 ± 79.3 in the VCt group, and 156.0 ± 25.2 in the VCb group, with a significantly higher number of falls in the VCt group compared to the other two experimental groups (** *p* < 0.01). These results suggest that the VCt protocol was the most challenging training protocol.

### 3.2. Effects of Different Exercise Protocols on Synaptic Plasticity

The effects of the three types of aerobic training on synaptic plasticity were analyzed in the CA1 region of mice hippocampal slices, and are shown in Figure 2. The CA1 subfield is one part of the three synaptic circuit characteristics of the hippocampal formation. It receives region information through the CA3 from the entorhinal cortex, and specifically directs the output signals towards the entorhinal cortex itself. In the CTRL mouse slices, inhibition of the expression of both PTP and LTP was observed, while the LTP maintenance phase was not affected. We observed that the PCt protocol did not improve synaptic plasticity, with respect to the sedentary group, since an impairment of the LTP induction phase was still present. The VCt protocol was even more stressful, since we observed a total inhibition of both the LTP induction and maintenance phase compared to the other experimental groups. On the contrary, the VCb protocol seemed to positively modulate synaptic plasticity. In fact, in the LTP induction phase, we observed that the PS values were similar to those of the CTRL and PCt groups, while in the LTP maintenance phase the PS values were significantly higher compared to those of the other three experimental groups.

The PS values recorded for each group of mice at various times after tetanic stimulation are reported in Table 1, where the values of significance are also shown.

### 3.3. Ultrastructural Analysis of the Muscle Tissues of NPC1+/− Sedentary Control and NPC1 +/− Trained Mice

To evaluate the adaptation of muscular plasticity following training with each protocol, ultrastructural analysis of muscle tissues was performed. The muscle tissue of the control group (Figure 3A) showed moderate misalignment of sarcomeric structures and the presence of numerous lipid droplets. The muscle fibrocells showed a correct organization with scarce localized vacuolizations between the myofibers. The mitochondria were well-preserved, but a dilation of the sarcoplasmic reticulum was observed. In the muscle tissue of mice trained with the PC protocol (Figure 3B), we observed an improvement in cellular conditions, with poor vacuolization and perfectly aligned myofibrils. The sarcomeres were well-organized, while the mitochondria were lined up along the fibers and in a perinuclear position. Furthermore, the nuclei were located along the cell periphery. The ultrastructural analysis of the muscle tissue from mice trained with the VC protocol (Figure 3C) showed a well-preserved tissue organization, with the correct orientation of the myofibrils. The mitochondria were well-preserved and were arranged variously between the fibers. Electro-bright spaces were present between the fibers, probably caused by a decrease in glycogen accumulations or by the presence of a mild cellular edema. We also observed the presence of scarce lipid droplets and a slight vacuolization.

## 4. Discussion

We analyzed the effects of different types of training on synaptic plasticity and muscle ultrastructural organization of heterozygous NPC1 mice. For these purposes, field potential recording on hippocampal slices and transmission electron microscopy analysis of anterior leg muscle were employed.

In a previous paper, we demonstrated that the administration of a short-term Uniform protocol consisting of nine training sessions in a mouse model of Niemann-Pick type C disease, a model of minimal Alzheimer’s Disease, was able to improve LTP impairment and counteract muscular deterioration in homozygous mice [10]. Since this type of protocol was not efficacy in the heterozygous NPC1 mice, probably because it was too weak and not demanding enough for this group, we decided to train the animals for a longer period (12 weeks) using more challenging protocols. We administered the PC and the VC training three times per week and the VC protocol two times per week. Our results showed that even if all three types of training protocols induced a weight increase not observed in the sedentary control, probably due to an augmentation of the lean mass, the rescue of synaptic plasticity, consisting of an up-modulation of the maintenance phase of the LTP was observed only in the hippocampal slices from NPC1 +/− VCb-trained mice. The PC training was not efficient enough and did not increase the LTP like the Uniform protocol, as previously reported, while the VCt was too stressful inducing a strong inhibition of the synaptic plasticity. This observation is also supported by the number of falls recorded during the three months of training that were significantly higher in the VCt than in the other two protocols employed. It is well reported in the literature that corticosteroids, particularly cortisol and inflammatory cytokines like IL-6 and TNF-α, modulate in negative fashion synaptic plasticity. This induces, in some conditions at higher doses, a complete blockade of the potentiation as we observed in our results in the hippocampal slices from NPC1 +/− VCt-trained mice [23,24,25]. This latter protocol may be too demanding and may induce a condition of overtraining that activates a defensive response in the body by releasing corticosteroid hormones and inflammatory substances. Nevertheless, it is well reported in the literature that physical exercise exerts several beneficial effects on cognitive and brain plasticity [26,27] by promoting hippocampal neurogenesis and enhancement in neuroplasticity, improving spatial memory and learning [28].

The observation of muscle tissue indicates that all three protocols are able to induce the recovery of the morphological structure: Improvement of cellular conditions, a reduction of vacuolization and maintenance of correct orientation of the myofibrils with respect to sedentary controls. In particular, the mitochondria are well-preserved, the sarcomeres are well-organized, and scarce lipid droplets are noted.

## 5. Conclusions

Interestingly, specific long-term motor training programs can induce adaptive changes both in the muscle and in the potential of hippocampal circuitry leading to long-term excitability changes, thereby opening a promising scenario for treatment and prevention of neurodegenerative diseases. In particular, the results of our study confirm the beneficial effects of exercise on quality of life, supporting the hypothesis that physical activity may represent an alternative therapeutic strategy for patients with Niemann-Pick type C disease. In fact, since there is no real cure for this pathology in humans to date, it may be useful to resort to new therapeutic, non-pharmacological and assistance approaches. Such as, for example, an adapted physical activity—which, if used in association with the current drug therapy, could be effective in reducing the progression and course of the disease.

## Figures and Tables

**Figure 1 jfmk-05-00053-f001:**
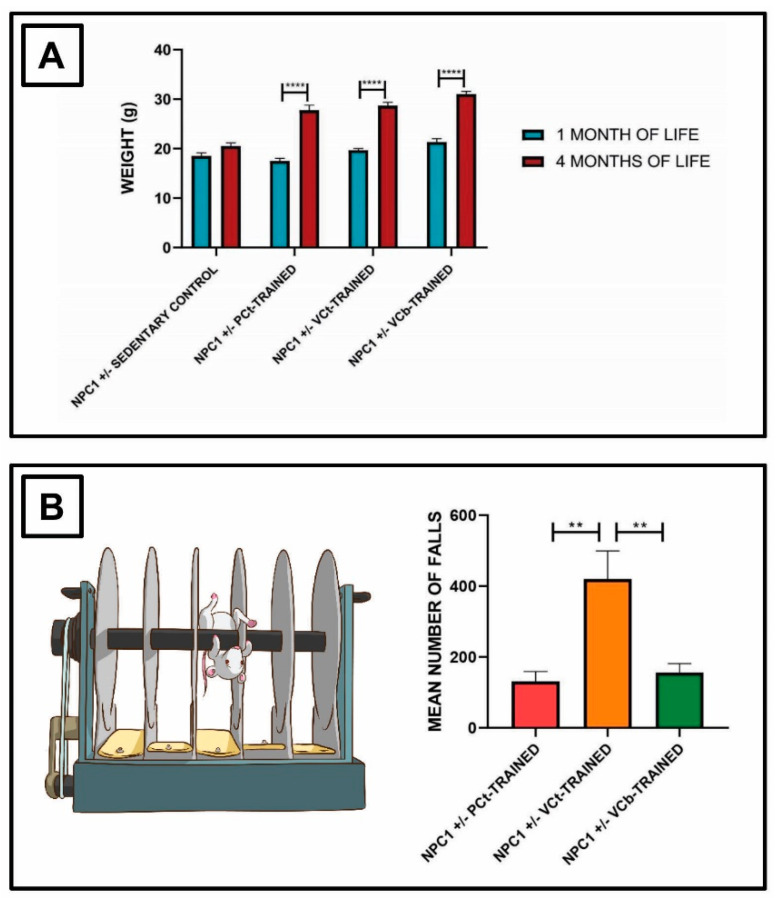
Effects of three different aerobic training protocols on physical and performance parameters. (**A**) The mean body weight in the control group was 18.5 ± 0.4 g at one month of life and 20.5 ± 0.4 g at four months of life. Regarding the trained mice, the mean body weight was 17.5 ± 0.5 g in the PCt group, 19.7 ± 0.3 g in the VCt group and 21.3 ± 0.7 g in the VCb group at one month of life. After training sessions, the mean body weight was 27.7 ± 1.0 g in the PCt group, 28.7 ± 0.7 g in the VCt group and 31.0 ± 0.6 g in the VCb group (**** *p* < 0.0001); (**B**) mean number of falls during the training protocol was 131.2 ± 27.7 in the PCt group, 420.0 ± 79.3 in the VCt group, and 156.0 ± 25.2 in the VCb group, with a significantly higher number of falls in the VCt group compared to the other two experimental groups (** *p* < 0.01).

**Figure 2 jfmk-05-00053-f002:**
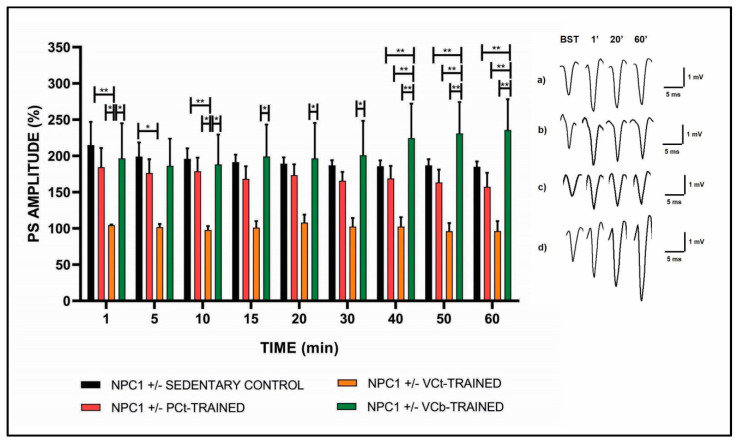
Synaptic plasticity in CA1 hippocampal subfield of NPC1+/− mice. The % PS amplitude after tetanic stimulation (HFS) as a function of time is shown in NPC1+/− SEDENTARY CONTROL (black bar, *n* = 5), in NPC1 +/− PCt-TRAINED (pink bar, *n* = 5), in NPC1 +/− VCt-TRAINED (yellow bar, *n* = 4) and in NPC1 +/− VCb-TRAINED (green bar, *n* = 4) mice slices at min 1, 5, 10, 15, 20, 30, 40, 50, 60. Bars in the plot are means ± SEM of values obtained from different slices. The insert shows recordings obtained from slices of NPC1+/− SEDENTARY CONTROL (**a**) NPC1 +/− PCt-TRAINED (**b**) NPC1 +/− VCt-trained (**c**) and NPC1 +/− VCb-trained (**d**) mice. Note that a significant statistical difference was reported between the experimental groups (Min 1: CTRL vs. VCt, ** *p* < 0.01; PCt vs. VCt and VCt vs. VCb, * *p* < 0.05. Min 5: CTRL vs. VCt, * *p* < 0.05. Min 10: CTRL vs. VCt, ** *p* < 0.01; PCt vs. VCt and VCt vs. VCb, * *p* < 0.05. Min 15, 20 and 30: VCt vs. VCb, * *p* < 0.05. Min 40, 50 and 60: CTRL vs. VCb, PCt vs. VCb and VCt vs. VCb, ** *p* < 0.01). The first curve of each group refers to the BST and was recorded before the application of the HFS, while the other curves refer to population spikes at times 1, 20 and 60 min after the HFS.

**Figure 3 jfmk-05-00053-f003:**
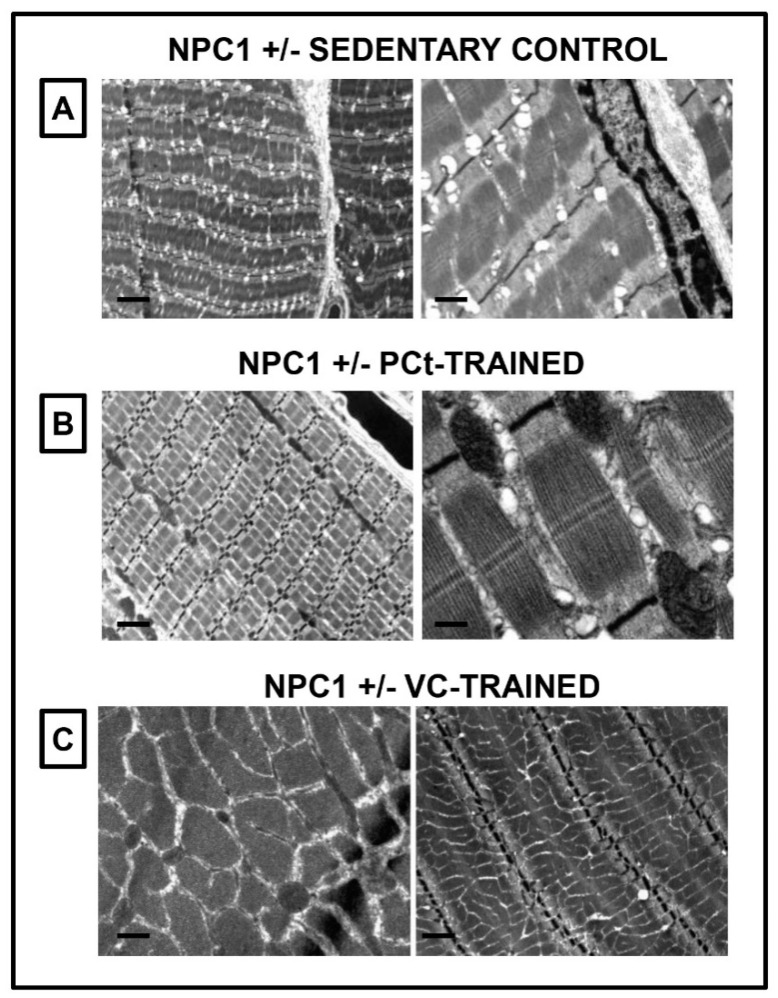
Muscular plasticity adaptation following training with the different protocols. (**A**) The ultrastructural analysis of the muscle tissue of NPC1 +/− SEDENTARY CONTROL mice showed moderate misalignment of the sarcomeres and scarce localized vacuolizations between the myofibers. The mitochondria were well-preserved, and a slight dilation of the sarcoplasmic reticulum was visible; (**B**) The ultrastructural analysis of the muscle tissue of NPC1 +/− PCt-TRAINED mice displayed an improvement in cellular conditions, with poor vacuolization and perfectly aligned myofibrils. The sarcomeres were well-organized, and the mitochondria were lined up along the fibers and in a perinuclear position; (**C**) In the muscle tissue of NPC1 +/− VC-TRAINED mice, a well-preserved tissue organization was visible, with the correct orientation of the myofibrils. The mitochondria were well-preserved and were arranged variously between the fibers. A presence of scarce lipid droplets and a slight vacuolization was visible. Scale bar, 1 µm.

**Table 1 jfmk-05-00053-t001:** Percentage of PS amplitude values recorded in the CA1 region of hippocampal slices from NPC1+/− mice at different times.

TIME (Min)	NPC1 +/- SEDENTARY CONTROL (% PS Amplitude)	NPC1 +/− CPt- TRAINED (% PS Amplitude)	NPC1 +/− CVt- TRAINED (% PS Amplitude)	NPC1 +/− CVb- TRAINED (% PS Amplitude)	SIGNIFICANCE
**1**	214.9 ± 32.0	184.5 ± 26.4	104.5 ± 0.9	196.6 ± 58.6	< 0.01
**5**	199.1 ± 19.5	176.4 ± 19.1	101.8 ± 4.3	186.4 ± 47.5	< 0.05
**10**	195.9 ± 14.6	178.8 ± 18.7	97.7 ± 5.5	188.2 ± 41.2	< 0.05
**15**	191.4 ± 10.3	168.5 ± 17.2	101.1 ± 8.9	199.2 ± 51.1	< 0.05
**20**	189.3 ± 8.8	173.5 ± 14.8	107.9 ± 14.1	196.6 ± 48.8	< 0.05
**30**	187.0 ± 7.0	165.7 ± 12.4	102.6 ± 11.5	200.9 ± 47.5	< 0.05
**40**	185.7 ± 8.2	168.9 ± 17.2	102.6 ± 12.9	224.5 ± 67.6	< 0.01
**50**	186.9 ± 8.5	163.4 ± 17.6	96.0 ± 11.4	231.0 ± 73.2	< 0.01
**60**	185.2 ±17.2	157.3 ± 19.5	96.3 ± 13.7	235.8 ± 77.4	< 0.01

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
