# Peer review of "Effects of Different Continuous Aerobic Training Protocols in a Heterozygous Mouse Model of Niemann-Pick Type C Disease"

_jfmk, 2020, doi:10.3390/jfmk5030053_

Round 1

Reviewer 1 Report

This manuscript describes experiments to evaluate the beneficial effects of two different types of aerobic training on hippocampal neuroplasticity in NPC1 +/- mice.

The manuscript is adequately designed with appropriate methodology and statistical analysis. The quality of data is sufficient and consistent with the conclusions.

Minor aspects need to be addressed:

  • Include a brief description of CA1 hippocampal subfield
  • In the main text define PS
  • Lanes 122-125 indicate the unit measure
  • Indicate the p value in the legend of figure 2

Author Response

We would like to thank the Reviewer for their comments on how to improve the manuscript, which has been revised accordingly.

Point 1: Include a brief description of CA1 hippocampal subfield.

Response 1: Thank you for this pointing out. As suggested by reviewer, we added a brief description of CA1 hippocampal subfield in lines 150-153: “The CA1 subfield is one part of the three synaptic circuit characteristic of the hippocampal formation that receives through the CA3 region informations from the enthorinal cortex, and specifically directs the output signals towards the enthorinal cortex itself”.

Point 2: In the main text define PS.

Response 2: Thank you for this pointing out. In line 104, we have defined PS: “population spike (PS), that indicates the electrical activity of a population of neurons”.

Point 3: Lanes 122-125 indicate the unit measure.

Response 3: Thank you for this pointing out. As recommended, we have added the unit measure in lines 123-127: "(CTRL 18.5 ± 0.4 g, PCt 17.5± 0.5 g, VCt 19.7 ± 0.3 g, VCb 21.3 ± 0.7 g)”; “(PCt 27.7 ± 1.0 g, VCt 28.7 ± 0.7 g, VCb 31.0 ± 0.6 g, ****p<0.0001)”; (20.5 ± 0.4 g)”.

Point 4: Indicate the p value in the legend of figure 2.

Response 4: Thank you for this pointing out. As suggested by reviewer, we have indicated the p value in the legend of figure 2, lines 168-172: “Note that a significant statistical difference was reported between the experimental groups (Min 1: PCt vs VCt and VCt vs VCb, *p<0.05; PCt vs VCt, **p<0.01. Min 5: CTRL vs VCt, *p<0.05. Min 10: PCt vs VCt and VCt vs VCb, *p<0.05; PCt vs VCt, *p<0.05. Min 15, 20 and 30: VCt vs VCb, *p<0.05. Min 40, 50 and 60: CTRL vs VCb, **p<0.01; PCt vs VCb, **p<0.01; VCt vs VCb, **p<0.01).”

Reviewer 2 Report

The manuscript: „Effects of Different Continuous Aerobic Training Protocols in a Heterozygous Mouse Model of Niemann-Pick type C Disease „ by Ida Cariati and colleagues demonstrate the beneficial effects of exercise on quality of life, in Niemann-Pick type C disease which might be helpful in designing therapeutic strategies in this disease. The experiments are nicely conducted and the results are well interpreted.

After thoroughly going through the manuscript, I have a couple of minor comments:

  1. The authors mention that they used mice carrying the genetic mutation for NPC1. Elaboration on genotype (exactly which mutation/s) is suggested as various mutations resulting in Niemann-Pick type C Disease are reported in NPC1 gene.
  2. A brief discussion on how the positive results on mice on laboratory setting could be implemented in designing therapeutic strategies in patients with Niemann-Pick type C Disease.

Author Response

We would like to thank the Reviewer for their comments on how to improve the manuscript, which has been revised accordingly.

Point 1: The authors mention that they used mice carrying the genetic mutation for NPC1. Elaboration on genotype (exactly which mutation/s) is suggested as various mutations resulting in Niemann-Pick type C Disease are reported in NPC1 gene.

Response 1: Thank you for this pointing out. As suggested by reviewer, we specified the genetic mutation for NPC1 of the heterozygous mice used in this experimental study in lines 74-75: “BALB/cNctr-Npc1m1N/J”.

Point 2: A brief discussion on how the positive results on mice on laboratory setting could be implemented in designing therapeutic strategies in patients with Niemann-Pick type C Disease.

Response 2: Thank you for this pointing out. As recommended, we added a brief description on how the positive results on mice on laboratory setting could be implemented in designing therapeutic strategies in patients with Niemann-Pick type C Disease in lines 247-253: “In particular, the results of our study confirm the beneficial effects of exercise on quality of life, supporting the hypothesis that physical activity may represent an alternative therapeutic strategy for patients with Niemann-Pick type C disease. In fact, since to date there is no real cure for this pathology in humans, it may be useful to resort to new therapeutic, non-pharmacological and assistance approaches, such as for example an adapted physical activity, which if used in association with the current drug therapy, could be effective in reducing the progression and course of the disease.”